# Cascade Speculative Drafting for Even Faster LLM Inference

**Ziyi Chen**   **Xiaocong Yang**   **Jiacheng Lin**   **Chenkai Sun**
**Kevin Chen-Chuan Chang**   **Jie Huang**
University of Illinois at Urbana-Champaign
`{ziyic2, kcchang, jeffhj}@illinois.edu`

## Abstract

Introduced to enhance the efficiency of large language model (LLM) inference, speculative decoding operates by having a smaller model generate a draft. A larger target model then reviews this draft to align with its output, and any acceptance by the target model results in a reduction of the number of the target model runs, ultimately improving efficiency. However, the drafting process in speculative decoding includes slow autoregressive generation and allocates equal time to generating tokens, irrespective of their importance. These inefficiencies collectively contribute to the suboptimal performance of speculative decoding. To further improve LLM inference, we introduce Cascade Speculative Drafting (CS Drafting), a speculative execution algorithm that incorporates two types of cascades. The *Vertical Cascade* eliminates autoregressive generation from neural models, while the *Horizontal Cascade* optimizes time allocation in drafting for improved efficiency. Combining both cascades, CS Drafting achieves greater speedup compared to the baselines in our experiments, while preserving the same output distribution as the target model.[1]

## 1 Introduction

The advent of Large Language Models (LLMs), like GPT-4 [17], has marked a significant milestone in the field of natural language processing (NLP). These models have not only excelled in various NLP tasks but have also found widespread applications in user-interactive settings, such as chatbots and virtual assistants. However, these applications involve an extremely high number of users, up to hundreds of millions daily. To serve in real-time at this scale, a low-latency system is not only cost-saving but also crucial for keeping the service running. In addition, the sheer scale of the service means that even a slight improvement in the latency of LLMs can greatly contribute to both the service provider and the community. Consequently, optimizing the latency of LLMs has become a critical area of research.

Unfortunately, the ever-growing size of LLMs significantly increases the latency, especially in long-form generation, as autoregressive LLMs generate tokens one by one. An emerging solution, known as speculative decoding [14, 4, 25], shows potential to mitigate this issue. In speculative decoding, a draft model (which is smaller and faster) generates $k$ tokens in each step (with $k$ being a hyperparameter) autoregressively, and these tokens are then reviewed by a target model (which is larger and slower) in parallel. In one single run, the target model will accept any tokens aligned with its output and further generate one token. The drafting process in speculative decoding enables the target model to generate multiple tokens in a single run while maintaining its output distribution unchanged. With a properly sized draft model, speculative decoding achieves a speedup of 2 to 3 times, making it a potential method for solving high latency issues.

---

[1]Code is publicly available at `https://github.com/lfsszd/CS-Drafting`.

38th Conference on Neural Information Processing Systems (NeurIPS 2024).

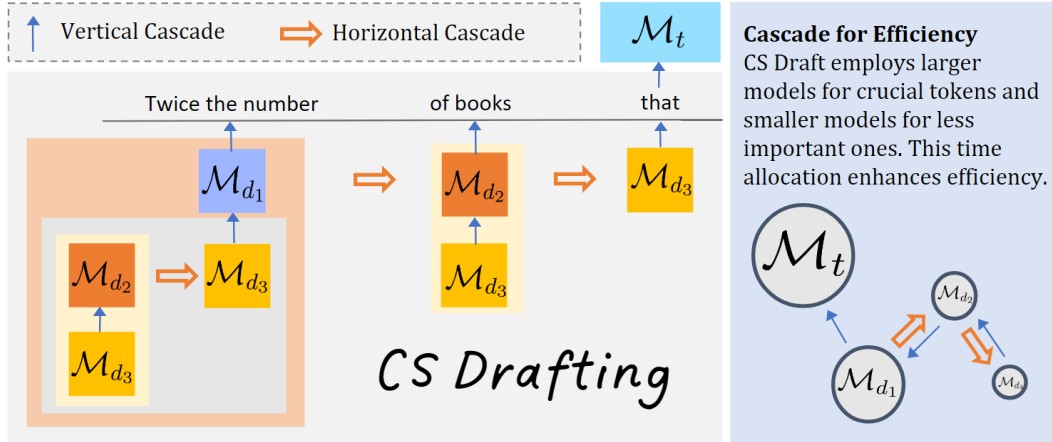

Figure 1: The CS Drafting algorithm features a recursive and resource-efficient design, implemented through two cascades: the horizontal cascade and the vertical cascade. The horizontal cascade involves using larger draft models to generate the earlier tokens and smaller models for the later tokens. The vertical cascade requires each model to review drafts from smaller models with the exception of the smallest model, which is a statistical language model. As the horizontal cascade and vertical cascade are orthogonal, CS Drafting combines both approaches for optimal efficiency. The figure shows an example of Cascade Speculative Drafting with target model $\mathcal{M}_t$ and draft models $\mathcal{M}_{d_1}$, $\mathcal{M}_{d_2}$, and $\mathcal{M}_{d_3}$.

However, since draft models are typically required to generate multiple tokens in multiple steps, where each generation still involves inefficient autoregressive decoding, the performance of speculative decoding could be limited by the drafting latency. This inefficiency is also indicated by Leviathan *et al.* [14], where it was observed that very small models (e.g., around two orders of magnitude smaller than the target model) are usually the best choice for drafting because their inference cost is lower compared to that of a larger draft model, despite the fact that larger draft models usually have higher-quality generation. This underscores that improving drafting efficiency is crucial for further enhancing the performance of speculative decoding.

In light of this, one key strategy to address this bottleneck is to avoid the inefficient autoregressive generation of neural draft models. Based on this consideration, it is noted that statistical language models, such as bigram language models, incur negligible latency and computational resource costs compared to neural language models, owing to their simple structure. However, because the tokens generated by statistical language models usually do not have a high probability of being accepted by the target model, speculative decoding with statistical language models alone may not yield optimal results compared to using a well-sized neural language model from the same family as the draft model. Nonetheless, we notice that it is not necessary to use only one draft model in speculative decoding—statistical language models can serve as the "draft" model for the neural draft model, thereby eliminating autoregressive generation from the neural draft model.

Furthermore, our analysis in Figure 2 reveals a pattern during the drafting step: tokens generated later in the sequence by the draft model show a progressively lower probability of being accepted by the target model. This is because the probability of a token being accepted is conditioned on the acceptance of the previous tokens. It indicates that later tokens from draft models are more prone to rejection, contributing less to the expected number of accepted tokens per draft step, yet incurring the same latency.

Inspired by the above observations, we propose **Cascade Speculative Drafting (CS Drafting)**, a speculative execution algorithm that comprises multiple draft models, with the smallest being a statistical language model. Each neural draft model reviews generations from a smaller model and then proposes its reviewed content to either a larger draft model or the target model. In this design, the drafting of each neural model is accelerated by drafting from a smaller model, avoiding the inefficiency of autoregressive generation from neural models. We refer to this tiered speculative decoding approach as the *Vertical Cascade*. In addition, we suggest the use of smaller, faster draft models for generating high-rejection tokens that are trailing in drafting generation, forming the

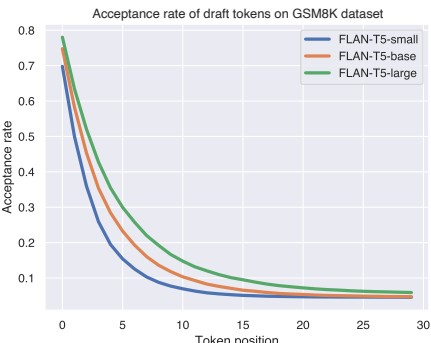
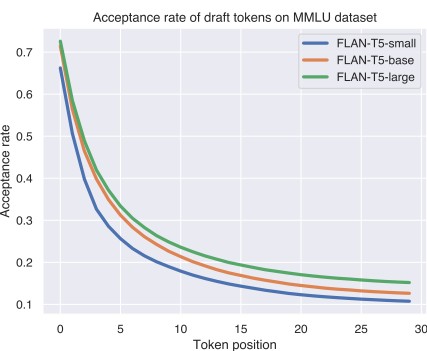

Figure 2: The probability of acceptance of draft tokens in relation to their positions in a single step of speculative decoding, evaluated on FLAN-T5-SMALL, BASE, and LARGE models on GSM8K and MMLU. The draft model generates 30 tokens at each step.

*Horizontal Cascade*. Along with the aforementioned *Vertical Cascade*, these strategies compose our complete CS Drafting approach, as illustrated in Figure 1.

Through theoretical analysis and empirical studies, we demonstrate that the CS Drafting algorithm outperforms speculative decoding in terms of latency across various tasks and settings, achieving an additional speedup of up to 81% over speculative decoding. These findings highlight the practical advantages and efficiency enhancements offered by both vertical and horizontal cascades.

The main contributions are summarized as follows:

- We introduce Cascade Speculative Drafting (CS Drafting), a speculative-execution-based algorithm that improves language model inference speed without sacrificing generation quality.
- We provide theoretical analyses supporting the effectiveness of the proposed CS Drafting approach.
- We conduct empirical experiments showing that CS Drafting achieves further speedup over speculative decoding across different tasks and settings.

## 2   Preliminary

The core concept of speculative decoding [14] involves the utilization of a small draft model for sequential token generation with validation by a larger target model resulting in reduced latency. This design accelerates sampling from autoregressive models without altering output distributions. At its heart, there are two key observations: 1) certain generations in language modeling are simpler than others and can be predicted by more efficient models correctly, and 2) using speculative execution along with a new sampling method enables faster, exact decoding from large models.

Specifically, let $x$ be the input tokens at a run and $\mathcal{M}_t$ and $\mathcal{M}_d$ are the target and the draft model respectively, $k$ be the number of draft tokens generated per step, and $\mathcal{M}_t(x)[i]$ and $\mathcal{M}_d(x)[i]$ be their probability output at $i$-th token when input is $x$. We interpret speculative sampling as a two-stage operation. In the proposing stage, we sample $\{x_{t+1}, ..., x_{t+k}\}$ from draft model $\mathcal{M}_d$ autoregressively and append them to $x$. In the reviewing stage, let $x_i \in \{x_{t+1}, ..., x_{t+k}\}$ represents the token at the current position, and we accept it if $\mathcal{M}_d(x)[i-1] \leq \mathcal{M}_t(x)[i-1]$; in the event that $\mathcal{M}_d(x)[i-1] > \mathcal{M}_t(x)[i-1]$, we reject $x_i$ with a probability of $1 - \frac{\mathcal{M}_t(x)[i-1]}{\mathcal{M}_d(x)[i-1]}$ and proceed to resample $x_i$ from a recalibrated distribution $norm(\max(0, \mathcal{M}_t(x)[i-1] - \mathcal{M}_d(x)[i-1]))$ and reject any token following $x_i$. At the end, the target model will generate one additional token following the accepted tokens. Such a design guarantees the output is the same as sampling autoregressively using the target model alone [14].

Speculative decoding was empirically validated on various tasks and model sizes, demonstrating a significant acceleration in inference times (2x-3x faster) compared to standard implementations, without affecting the outputs. Importantly, it does not require task-specific training, altering model architectures, or changing training procedures, making it a practical solution for reducing the latency of LLM inference.

---
**Algorithm 1** CascadeSpeculativeDraftingStep
---
**Require:** draft models $\{\mathcal{M}_{d_1}, ..., \mathcal{M}_{d_n}\}$, target mode $\mathcal{M}_t$, $prefix$, flag $isFirstCall$, hyperparameters $K_{nn}$, $l$

    draftList $\leftarrow [\mathcal{M}_{d_1}, ..., \mathcal{M}_{d_n}]$
    ▷ Initialize curGen and curProb.
    curGen $\leftarrow prefix$, curProbs $\leftarrow$ a list of ones with the same length as prefix
    ▷ Unpack the a list of $k$ for the current function call.
    $[k_1, ..., k_{n-1}] \leftarrow$ first row of $K_{nn}$
    ▷ Generate using MaG for the Base case of the recursive call.
    **if** draftList is empty **then**
        $\mathcal{M} \leftarrow$ first element of draftList
        $res \leftarrow \mathcal{M}(curGen)$
        **return** $res.generation, res.logits$
    **end if**
    ▷ Perform the horizontal cascade with the for loop.
    **for** $i \leftarrow 1$ **to** $n$ **do**
        ▷ Prepare the arguments for the next recursive call.
        curTarget $\leftarrow$ the $i$-th item of draftList
        curDraftList $\leftarrow$ the sublist of draftList starting from index $i + 1$
        curK $\leftarrow$ the submatrix of $K_{nn}$ from with the top-left corner at $(i + 1, i + 1)$ extending to the bottom-right corner curPrefix $\leftarrow$ curGen
        **while** curGen.length - curPrefix.length is less than $k_i$ **do**
            curPrefix $\leftarrow$ curGen
            ▷ Perform the vertical cascade with the recursive call.
            $[x_1, .., x_u], [p_1, p_2, ..., p_v] \leftarrow$ *CascadeSpeculativeDraftingStep*(curDraftList, curTarget, curPrefix, False, curK, $l$)
            curGen $\leftarrow [x_1, .., x_u]$
            s $\leftarrow$ curProbs.length + 1
            Add elements of $[p_1, p_2, ..., p_v]$ to curProbs
        **end while**
    **end for**
    ▷ Set lenience to 1 when the original target model reviews.
    **if** isFirstCall **then**
        $l \leftarrow 1$
    **end if**
    ▷ Use $\mathcal{M}_t$ to review the draft generation.
    $[x_1, ..., x_{out}], [p'_1, p'_2, ..., p'_{out}] =$ review($\mathcal{M}_t$, curGen, curProbs, l)
    **return** $[x_1, ..., x_{out}], [p'_1, p'_2, ..., p'_{out}]$
---

# 3 Cascade Speculative Drafting

In this section, we introduce our proposed method, Cascade Speculative Drafting (CS Drafting), a speculative execution algorithm that incorporates two types of cascades: *vertical cascade* and *horizontal cascade*.

## 3.1 Vertical Cascade

A notable inefficiency of the speculative decoding algorithm is the reliance on the autoregressive generation of a smaller draft model. Since the draft model must run $k$ times for each target model run, the cost can still be significant despite its smaller size. In light of this, we reduce the drafting inefficiency by using an even smaller model to assist in drafting and employing the original draft model to review the generation of this smaller model. In addition, since this process can be performed again on the draft model that drafts for the original model, we recursively perform this process until it reaches a statistical draft model that involves negligent cost, such as a bigram language model. In this approach, we expect each recursion step will reduce the drafting latency without altering the output distribution. We refer to this recursive speculative approach as *Vertical Cascade*.

Additionally, we incorporate *lenience*, a hyperparameter that loosens the review process by the target model, allowing for faster speed at the trade-off of potentially differing results from the target model [14]. Lenience can be adopted during sampling or greedy decoding with speculative decoding. Let lenience $l \in [1, \infty)$. When sampling, the acceptance condition for token $x_i$ is transformed to $\mathcal{M}_d(x)[i] \leq l \times \mathcal{M}_t(x)[i]$. If the acceptance condition is not satisfied, with a probability of $1 - \frac{l \times \mathcal{M}_t(x)}{\mathcal{M}_d(x)}$, we reject $x_i$ and any following tokens.[2] When performing greedy decoding, the acceptance condition becomes deterministic and is simply either $argmax\mathcal{M}_d(x)[i] = argmax\mathcal{M}_t(x)[i]$ or $\mathcal{M}_d(x)[i] \leq l \times \mathcal{M}_t(x)[i]$.

For the speculative decoding algorithm, the reduced quality introduced by lenience is generally undesirable. However, for the vertical cascade approach, lenience affects the final output only if it is applied when the target model reviews. Therefore, we can limit the application of lenience in the vertical cascade only when draft models review and do not apply lenience when the target model reviews. This can ensure the final output is not altered while further reducing latency.

## 3.2 Horizontal Cascade

Another key observation is that during the drafting steps of speculative decoding, not all drafting tokens are created equal, as illustrated in Figure 2. The first draft token is more likely to be accepted as it only depends on itself; the last token is rarely accepted, as it has a chance of being reviewed only if all preceding tokens are accepted. From a theoretical perspective, assume the event of acceptance of each token being a Bernoulli distribution with probably $p$, the probability of $n$-th token being accepted is $p^n$, implying an exponential decrease of value for tokens generated later in the sequence.

Inspired by this observation, we designed *Horizontal Cascade*, an approach that improves time allocation by draft token allocation. Horizontal Cascade assigns the largest draft model to perform the generation of the first draft token due to its highest output alignment with the target model, and it progressively uses a smaller as the new draft token to be generated is less likely to be accepted. This process stops after the smallest model, i.e., a statistical language model finishes. This design reduces the time cost of generating unimportant draft tokens with a costly draft model, leading to a reduction in overall latency.

## 3.3 Max-Gram for Better Statistical Drafting

As both Vertical Cascade and Horizontal Cascade remark cascade toward faster draft models, a statistical language model, which is the basis of the cascade, becomes essential for the efficiency of both approaches. In our pursuit of a more effective statistical language model, we noticed a general pattern: in language model generation, some words and phrases from the input query frequently reappear in the generated content. In light of this observation, we designed the **Ma**x-**G**ram (MaG) algorithm. It greedily identifies maximal matches between the initial input (or existing generation) and tokens from the end of the generation. In cases where there is no match, we resort to a bigram model based on the probability distribution of Wikipedia (chosen to maintain the generality). We include a GPU-friendly version of the Max-Gram algorithm in Appendix A.

## 3.4 Algorithm

Combining the horizontal and vertical cascades, the algorithm of cascade speculative decoding is presented in Algorithm 1. At its center, the horizontal cascade is realized by the for loop, while the vertical cascade is implemented through recursive calls. Notably, the MaG model is incorporated as the smallest draft model to avoid autoregressive generation from a neural model. An example of CS Drafting is shown in Figure 1.

The algorithm requires an upper-triangular hyperparameter, $K_{nn}$, with each row serving as the stop criteria for a layer of recursive calls. For simplicity, we assume the lenience $l$ is universal for the algorithm, except when the target model is under review; thus, the algorithm can benefit from the speedup of lenience without altering the output distribution.

---

[2]For simplicity, we assume the probability outputs are not standardized. We refer the readers to the speculative decoding paper [14] for the discussion on standardized sampling.

## 4 Analysis

In this section, we provide theoretical analyses for Cascade Speculative Drafting. We begin with some notions. Let $\mathcal{M}_t$ be the target model, $\mathcal{M}_d$ be the draft model, and $k$ be the number of draft tokens generated per step.

**Expected acceptance rate** $\alpha(\mathcal{M}_t, \mathcal{M}_d)$ is the probability of draft generation by $\mathcal{M}_d$ being accepted by target model $\mathcal{M}_t$.

**Cost coefficient** $c(\mathcal{M}_t, \mathcal{M}_d)$ is the ratio of time for a single run of draft model $\mathcal{M}_d$ over target model $\mathcal{M}_t$.

**Expected walltime improvement factor (EWIF)** is the expected time improvement achieved by an algorithm under the i.i.d. assumption of token acceptance.

Despite the simple setting of EWIF, it is demonstrated that it aligns with the experimental results in most instances[14]. Therefore, our analysis will concentrate on EWIF.

### 4.1 Vertical Cascade

We analyze EWIF of vertical cascade using generating functions, a well-studied topic in combinatorial mathematics [24]. The properties of generating functions are useful in the recursion and evaluation process making our final expression simple.

We begin with the derivation of the probability generating function for speculative decoding.

**Theorem 4.1.** *For speculative decoding between $\mathcal{M}_t$ and $\mathcal{M}_d$, let $p_i$ be the probability of generating $i$ tokens. The probability generating function of $p_i$ satisfies the following equation:*

$$\phi_{(\alpha,k)}(x) = 1 + (x-1)\frac{1 - \alpha^{k+1}x^{k+1}}{(1-\alpha x)}, \tag{1}$$

*where $\alpha = \alpha(\mathcal{M}_t, \mathcal{M}_d)$.*

Proof in Appendix C.1.

**Corollary 4.2.** *The EWIF of speculative decoding is $\frac{\phi'_{(\alpha,k)}(1)}{(ck+1)} = \frac{1-\alpha^{k+1}}{(1-\alpha)(ck+1)}$.*

We use the generating function to derive the EWIF of a vertical cascade and analyze the case involving two draft models, $\mathcal{M}_{d_1}$ and $\mathcal{M}_{d_2}$.

**Theorem 4.3.** *Assume $k$ to be the speculative decoding parameter between $\mathcal{M}_{d_1}$ and $\mathcal{M}_{d_2}$, and $n$ to be the number of steps $\mathcal{M}_t$ reviews. The EWIF by this system is*

$$\frac{1 - \alpha\phi^n(\alpha)}{(1-\alpha)(1 + nc_{d_1} + nkc_{d_2})}, \tag{2}$$

*where $\phi(x) = \phi_{(\alpha(\mathcal{M}_{d_1}, \mathcal{M}_{d_2}),k)}(x)$, $\alpha = \alpha(\mathcal{M}_t, \mathcal{M}_d)$, and $c_{d_1}, c_{d_2}$ be $c(\mathcal{M}_t, \mathcal{M}_{d_1}), c(\mathcal{M}_t, \mathcal{M}_{d_2})$ respectively.*

**Corollary 4.4.** *$\phi_{\alpha',k}(\alpha) < \alpha$ for any $1 > \alpha > 0, 1 > \alpha' > 0, k > 0$, so if $c_{d_2} \ll 1$, the EWIF of $\mathcal{M}_{d_1}$ and $\mathcal{M}_{d_2}$ is higher than EWIF of $\mathcal{M}_{d_1}$ alone.*

Proof in Appendix C.2.

Therefore, with the statistical model having negligible cost (i.e., $c_{d_2} \ll 1$), it can almost always improve the efficiency of an SD system.

### 4.2 Horizontal Cascade

We also present an analysis of the walltime improvement offered by the horizontal cascade.

To assist the analysis, we establish the notions. Let $\mathcal{M}_t$ be the target model, $\{\mathcal{M}_i\}$ be the draft models assisting generation with $\mathcal{M}_i$ being the draft model generating the $i$-th token. In the simpler case of the speculative decoding, $\mathcal{M}_i = \mathcal{M}_d$ for any $i$. Let $x$ be the input to the model at a single run, $\mathcal{M}_t(x)$ and $\mathcal{M}_i(x)$ are then the output probability distribution with input $x$. To simplify notation, let $\alpha_i = \alpha(\mathcal{M}_t(x), \mathcal{M}_i(x))$ and $c_i = c(\mathcal{M}_t(x), \mathcal{M}_i(x))$.

| Dataset | $\mathcal{M}_{d_1}$ | $\mathcal{M}_{d_2}$ | $k_1$ | $k_2$ | EWIF |
|---------|---------------------|---------------------|-------|-------|------|
| CNNDM | SMALL | - | 9 | - | 2.65 |
| CNNDM | BASE | - | 8 | - | 2.96 |
| CNNDM | BASE | SMALL | 5 | 3 | **3.03** |
| ENDE | SMALL | - | 12 | - | 3.61 |
| ENDE | BASE | - | 11 | - | 3.75 |
| ENDE | BASE | SMALL | 5 | 8 | **3.93** |

Table 1: Simulated EWIF under the assumption that the acceptance distribution is a Bernoulli distribution. BASE and SMALL refer to FLAN-T5-BASE and FLAN-T5-SMALL. In the simulation, speculative sampling with horizontal cascade exceeded the performance of the vanilla speculative decoding on both CNN Dailymail [16] and WMT EnDe [2] datasets.

**Theorem 4.5.** *The expected walltime improvement factor (EWIF) of the horizontal cascade is*
$$T(k, \alpha_1, ..., \alpha_k, c_1, ..., c_k) = \frac{\sum_{i=0}^{k} \prod_{j=1}^{i} \alpha_j}{1 + \sum_{i=1}^{k} c_i}.$$

Furthermore, theorem 4.5 can be used to analyze the importance of the tokens in the drafting step.

**Corollary 4.6.** *The probability of $i$-th token being accepted is $\prod_{j=1}^{i} \alpha_j$. The derivative of EWIF with respect to $\alpha_l$ is $\frac{dT(k, \alpha_1, ..., \alpha_k, c_1, ..., c_k)}{d\alpha_l} = \frac{\sum_{i=l}^{k} \prod_{j=1, j \neq l}^{i} \alpha_j}{1 + \sum_{i=1}^{k} c_i}$. Specifically, $\frac{dT(k, \alpha_1, ..., \alpha_k, c_1, ..., c_k)}{d\alpha_1} = \frac{\sum_{i=1}^{k} \prod_{j=2}^{i} \alpha_j}{1 + \sum_{i=1}^{k} c_i}$ and $\frac{dT(k, \alpha_1, ..., \alpha_k, c_1, ..., c_k)}{d\alpha_k} = \frac{\prod_{j=1}^{k-1} \alpha_j}{1 + \sum_{i=1}^{k} c_i}$.*

Using the information provided by Leviathan *et al.* [14], we calculate a simulated EWIF under the assumption that the event of acceptance by the target model is a Bernoulli trial. The results, shown in Table 1, indicate that speculative sampling with a horizontal cascade achieved better EWIF than vanilla speculative sampling under this assumption.

# 5 Experiments

## 5.1 Experimental Setup

**Metrics** We use both our proposed standardized walltime improvement and walltime for evaluation:

- **Standardized walltime improvement (SWI)** assumes each forward run of a model takes a constant amount of time which can be recorded data of previous work [14] or heuristics such as the total number of the parameters of a model. Under this assumption, the value of SWI is the speedup of the speculative method over autoregressive generation. SWI alleviates hardware variation and features full reproducibility of experiment results.

- **Walltime** refers to the actual elapsed time taken to complete a specific task or operation in a real-world scenario. Despite being less reproducible and sensitive to noise, walltime better represents the performance for individual users. In our experiment, walltime is measured in the form of the number of tokens generated per second on our GPU.

**Datasets** We chose two commonly used datasets for our experiments. For both datasets, we conducted experiments in a zero-shot chain-of-thought setup [13, 23]:

- **GSM8K** [7] is a dataset comprising 8,500 high-quality, linguistically diverse, grade-school math word problems. It focuses on multi-step reasoning with problems that are typically solvable using basic arithmetic in 2 to 8 steps.

- **MMLU** [10], or Massive Multitask Language Understanding, is a benchmark for testing how well large language models grasp knowledge. It encompasses 57 diverse subjects, ranging from elementary science to advanced law.

**Baselines** To verify the effectiveness of both vertical and horizontal cascade strategies intuitively, we first compare the performance of CS Drafting with different numbers of cascades, as well as its performance against standard speculative decoding [14]. Additionally, CS Drafting can also operate

| Dataset | Algorithm | $\{\mathcal{M}_{d_i}\}$ | Speedup (MS) | Speedup (PW) |
|---------|-----------|------------------------|--------------|--------------|
| GSM8K | Autoregressive | - | 1 | 1 |
| GSM8K | S Decoding | BASE | 3.38 | 2.99 |
| GSM8K | S Decoding | SMALL | 3.06 | 2.76 |
| GSM8K | CS Drafting | BASE, MAG | 3.70 | 3.27 |
| GSM8K | CS Drafting | SMALL, MAG | 3.19 | 2.82 |
| GSM8K | CS Drafting | BASE, SMALL, MAG | **3.88** | **3.43** |
| MMLU | Autoregressive | - | 1 | 1 |
| MMLU | S Decoding | BASE | 3.97 | 3.42 |
| MMLU | S Decoding | SMALL | 4.12 | 3.51 |
| MMLU | CS Drafting | BASE, MAG | 4.56 | 4.21 |
| MMLU | CS Drafting | SMALL, MAG | 4.39 | 3.99 |
| MMLU | CS Drafting | BASE, SMALL, MAG | **4.88** | **4.32** |

Table 2: The experimental results on FLAN-T5. Speedup (MS) is the standardized walltime improvement with the assumption that the latency of each run of a model is its number of parameters (model size). Speedup (PW) is the SWI with the assumption that the latency of each run of a model is the time cost data reported from previous work [14].

vertically by combining with other advanced decoding methods. To verify this, we also leverage tree attention, as used in Medusa [3], and compare its performance with Medusa.

**Implementation Details** To ensure the generality of our findings, we perform experiments on both encoder-decoder and decoder-only models. For encoder-decoder models, we choose our target and draft models from the FLAN-T5 [6] family for our experiment, as there is a large variation in model sizes within the FLAN-T5 family (ranging from 77 million to 11 billion parameters). We use FLAN-T5-XXL as our target model, FLAN-T5-BASE and FLAN-T5-SMALL as our reviewing draft models. For decoder-only models, we select Vicuna-7B [28], a fine-tuned version of LLaMA [20] as the target model. We use a 68M model with the same tokenizer as the reviewing draft model[3]. We also leverage tree attention [15, 3] with CS Drafting for the experiments on Vicuna-7B. In both cases, the Max-Gram algorithm is used as the generating draft model. Since we do not observe any significant difference between sampling with temperature 1 and greedy decoding in previous speculative decoding experiments [14], and to ensure our experiments are fully reproducible, we perform sampling at temperature 0, i.e., using greedy decoding by default. To align our experiment with current common usage, we do not perform fine-tuning for CS Drafting, and the generation is conducted in a zero-shot manner. We include hyperparameter details in Appendix B. All of our experiments involving walltime are performed on a single NVIDIA A40 GPU.

## 5.2 Experimental Results

Table 2 presents the main experimental results. In two settings of SWI, Cascade Speculative Drafting has outperformed the speculative decoding algorithm. For GSM8K, CS Drafting achieved a maximum additional speedup of 44% over the fastest speculative algorithm; for MMLU, the maximum additional speedup improvement over speculative decoding is 81%.

**Effectiveness of MaG** When comparing CS Drafting with one neural model and MaG against the fastest speculative decoding setup, we found that CS Drafting with one neural model gained up to a 70% speedup on MMLU and a 32% speedup on GSM8K. Notably, the MaG algorithm only involves a bigram model with parameters equal to the tokenizer size, making its memory cost negligible." In addition, the speedup gained using CS Drafting with one neural model involves no additional deployment overhead while reducing both latency and computational cost, making it a superior choice over speculative decoding.

**Draft Model Size** Despite FLAN-T5-SMALL mostly outperforming FLAN-T5-BASE as a draft model for speculative decoding, in CS Drafting with the aid of MaG, FLAN-T5-BASE consistently

---

[3]https://huggingface.co/double7/vicuna-68m

| Algorithm | GSM8K Walltime (Tokens/s) | MMLU Walltime (Tokens/s) |
|---|---|---|
| Autoregressive | 33.34 | 33.11 |
| S Decoding | 44.31 | 43.65 |
| CS Drafting | 56.72 | 55.60 |
| Medusa | 61.87 | 56.19 |
| CS Drafting + Tree Attention | **63.81** | **63.37** |

Table 3: The experimental results on Vicuna-7B.

| $K_{00}$ | Walltime (tokens/s) |
|---|---|
| 1 | 56.16 |
| 2 | 55.51 |
| 3 | 53.73 |

Table 4: Results on GSM8K with Vicuna-7B under different generation length limits.

outperforms FLAN-T5-SMALL. This implies that with the limitation of a single draft model, the ideal size of the draft model might increase with the assistance of the MaG model.

**Results of Decoder-only Models** As shown in Table 3, CS Drafting achieves a significant walltime improvement over speculative decoding. Moreover, CS Drafting exceeds Medusa when combined with tree attention [15, 3]. This suggests that CS Drafting can be integrated with other efficient designs for speculative decoding to further accelerate inference. We leave the exploration of more advanced combinations as future work.

**Ablation Study** We perform a hyperparameter study on $K_{00}$, the hyperparameter with the greatest effect on our experiments. As shown in Table 4, the performance of CS Drafting only decreases slightly when this hyperparameter is sub-optimal. Therefore, end users who do not require maximum performance can use a simple setup to achieve near-optimal performance with CS Drafting. Furthermore, we conduct an ablation study by removing the horizontal cascade. On GSM8K, with Vicuna-7B and $K_{00} = 1$, this results in a performance drop from 56.16 to 53.55.

## 6 Related Work

### 6.1 Efficienct Methods for Language Model Inference

In the era of large language models, efficiency during inference becomes a key to model service. To reduce the model inference cost and speed up, several efficient methods have been proposed, including pruning, knowledge distillation and quantization [21]. Model pruning takes structured [26, 22] or unstructured [9, 5] methods to remove the redundant model parameters to reduce the storage memory and increase inference speed. Knowledge distillation takes the approach of transferring knowledge from a superior teacher model to a smaller student model [11, 8]. Quantization maps high-precision data representations (e.g. 32 bits) into low-precision ones (e.g. 8 bits) to reduce memory consumption [1, 18].

### 6.2 Speculative Decoding

With the success of Speculative Decoding [4, 14] in reducing the large language model inference latency, some recent works have attempted to improve Speculative Decoding by reducing the rejection rate. Zhou *et al.* [29] propose using generalized knowledge distillation and achieve a lower rejection rate compared to other knowledge distillation methods. Avoiding an additional draft model, self-drafting is an approach to speculative decoding by reusing part of the target model together with added weight to perform drafting [27, 12]. Tree attention involves generating multiple candidates during drafting to increase the chance of acceptance [19, 15]. Besides reducing the rejection rate, improving drafting efficiency can also reduce latency. Spector *et al.* [19] propose using speculative decoding for drafting, showing similarities to the vertical cascade; however, their method only has two layers of speculative decoding and does not observe the recursive nature of the vertical cascade nor the lenience among draft models, two crucial aspects for the performance of vertical cascade.

# 7 Conclusion

In this work, we propose a novel algorithm, CS Drafting, which involves two cascades: the vertical cascade and the horizontal cascade. The vertical cascade eliminates the necessity of autoregressive generation from a neural language model, while the horizontal cascade effectively allocates the cost of drafting tokens at different positions. CS Drafting achieves additional speedup over baselines in various settings while maintaining the same output distribution as the target model.

# 8 Limitations

Our experiments demonstrate strong performance for our methods. However, it is possible, though unlikely, that the outcome might differ on a system with different hardware configurations. To account for this, we report both standardized walltime improvement and raw walltime for a more robust evaluation. Additionally, we provide theoretical analyses to justify the improvements, which is independent of the hardware configurations.

## Acknowledgement

This material is based upon work supported by the National Science Foundation IIS 16-19302 and IIS 16-33755, Zhejiang University ZJU Research 083650, IBM-Illinois Center for Cognitive Computing Systems Research (C3SR) and IBM-Illinois Discovery Accelerator Institute (IIDAI), grants from eBay and Microsoft Azure, UIUC OVCR CCIL Planning Grant 434S34, UIUC CSBS Small Grant 434C8U, and UIUC New Frontiers Initiative. Any opinions, findings, conclusions, or recommendations expressed in this publication are those of the author(s) and do not necessarily reflect the views of the funding agencies.

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

## A  Max-Gram Implementation

Listing 1: Max-Gram Algorithm

```python
def torch_index(t, value):
    return (t == value).nonzero(as_tuple=True)[0][0]

def max_gram(input_ids, encoder_ids, n=1):
    matches = (encoder_ids[0] == input_ids[0, -1]).int()
    if matches.sum() < 1:
        return None
    for i in range(2, input_ids.shape[-1] + 1):
        new_matches = (encoder_ids[0, :(-1 * (i - 1))] == input_ids[0, -1 * i]).int()
        combined_matches = (2 - new_matches == matches[1:]).int()
        if combined_matches.sum() < 1:
            index = torch_index(torch.cat(
                (
                    torch.tensor([0] * (i - 1), device=torch.device(encoder_ids.device)),
                    matches
                ),
                dim=-1
            ), 1)
            return encoder_ids[:, index:index + n]
        else:
            matches = combined_matches
    index = torch_index(torch.cat((
        torch.tensor([0] * (encoder_ids.shape[-1] - matches.shape[-1])), matches), dim=-1
    ), 1)
    return encoder_ids[:, index+1:index + n+1]
```

## B  Hyperparameters

To reduce the number of hyperparameters to tune, we use MaG to generate 10 tokens at once, as it is rare for more than 10 tokens to be accepted with the exception when CS Drafting is combined with tree attention. We do not use lenience when the reviewer is $\mathcal{M}_t$ to ensure the output distribution does not change. We also avoid lenience between MaG and its reviewer, since there is still a significant performance gap between MaG and a neural model. With these constraints, we are left with at most four hyperparameters: $k_{11}$, $k_{12}$, $k_{22}$, and $l$. For the CS Drafting step where the target model is the reviewer, $k_{11}$ and $k_{12}$ are used. $k_{21}$ and $l$ are used in the step where $\mathcal{M}_{d_1}$ is the reviewer. The results of experiments on encoder-decoder models with hyperparameters are shown in Table 5.

When performing experiments with the decoder-only model, we fixed the hyperparameters of CS Drafting for different datasets to better align with most users who do not perform hyperparameter tuning. The k-matrix for CS Drafting is $[[2, 10], [0, 10]]$. When adding tree attention, we limit it to only the lead node with the highest probability of having children; the k-matrix is $[[1, 3], [0, 1]]$ with the number of children for each leading node being 8, while the other nodes have no children.

## C  Proof

### C.1  Proof for Theorem 4.1

*Proof.* The probability of accepting $i$ tokens is $\alpha^i - \alpha^{i+1}$, with the exception of the $k+1$-th token, which has a probability of $\alpha^k$ of being accepted. This is because it requires all the first $i$ tokens to be accepted and the $i+1$-th token to be rejected for this to happen. Therefore,

$$\phi_{(\alpha,k)}(x) = \alpha^k x^{k+1} + \sum_{i=0}^{k-1} (\alpha^i - \alpha^{i+1}) x^{i+1}. \tag{3}$$

| Dataset | Algorithm | $\{\mathcal{M}_{d_i}\}$ | Speedup (MS) | $k_{11}$ | $k_{12}$ | $k_{22}$ | $l$ |
|---|---|---|---|---|---|---|---|
| GSM8K | Autoregressive | - | 1 | - | - | - | - |
| GSM8K | S Decoding | BASE | 3.38 | 10 | - | - | - |
| GSM8K | S Decoding | SMALL | 3.06 | 11 | - | - | - |
| GSM8K | CS Drafting | BASE, MAG | 3.70 | 10 | - | - | - |
| GSM8K | CS Drafting | SMALL, MAG | 3.19 | 11 | - | - | - |
| GSM8K | CS Drafting | BASE, SMALL, MAG | **3.88** | 8 | 13 | 1 | 3 |
| MMLU | Autoregressive | - | 1 | - | - | - | - |
| MMLU | S Decoding | BASE | 3.97 | 13 | - | - | - |
| MMLU | S Decoding | SMALL | 4.12 | 19 | - | - | - |
| MMLU | CS Drafting | BASE, MAG | 4.56 | 13 | - | - | - |
| MMLU | CS Drafting | SMALL, MAG | 4.39 | 14 | - | - | - |
| MMLU | CS Drafting | BASE, SMALL, MAG | **4.88** | 5 | 19 | 1 | 5 |
| Dataset | Algorithm | $\{\mathcal{M}_{d_i}\}$ | Speedup (PW) | $k_{11}$ | $k_{12}$ | $k_{22}$ | $l$ |
| GSM8K | Autoregressive | - | 1 | - | - | - | - |
| GSM8K | S Decoding | BASE | 2.99 | 8 | - | - | - |
| GSM8K | S Decoding | SMALL | 2.76 | 8 | - | - | - |
| GSM8K | CS Drafting | BASE, MAG | 3.27 | 9 | - | - | - |
| GSM8K | CS Drafting | SMALL, MAG | 2.82 | 11 | - | - | - |
| GSM8K | CS Drafting | BASE, SMALL, MAG | **3.43** | 5 | 9 | 1 | 3 |
| MMLU | Autoregressive | - | 1 | - | - | - | - |
| MMLU | S Decoding | BASE | 3.42 | 10 | - | - | - |
| MMLU | S Decoding | SMALL | 3.51 | 11 | - | - | - |
| MMLU | CS Drafting | BASE, MAG | 4.21 | 6 | - | - | - |
| MMLU | CS Drafting | SMALL, MAG | 3.99 | 13 | - | - | - |
| MMLU | CS Drafting | BASE, SMALL, MAG | **4.32** | 5 | 8 | 1 | 5 |

Table 5: The experimental results on FLAN-T5 with hyperparameter details. Speedup (MS) is the standardized walltime improvement with the assumption that the latency of each run of a model is its number of parameters (model size). Speedup (PW) is the SWI with the assumption that the latency of each run of a model is the time cost data reported from previous work [14]. $k_{11}$, $k_{12}$, $k_{22}$, $l$ are the hyperparameters. $k_{11}$ and $k_{12}$ represent the step limitation target model and the draft models, $k_{22}$ is the step limitations between the first and second draft model, and $l$ is lenience as shown in algorithm 1. For speculative decoding, the $k_{11}$ is simply the $k$.

By rearranging the terms, we can achieve an expression much easier to work with

$$\phi_{(\alpha,k)}(x) = x + \sum_{i=1}^{k} \alpha^i(x^{i+1} - x^i) \tag{4}$$

$$= x + (x-1)\sum_{i=1}^{k} \alpha^i(x^i) \tag{5}$$

$$= x + (x-1)\frac{1 - \alpha^{i+1}x^{i+1}}{1 - \alpha x}. \tag{6}$$

$\square$

## C.2 Proof for Theorem 4.3

*Proof.* Let $\alpha' = \alpha(\mathcal{M}_{d_1}, \mathcal{M}_{d_2})$. We first calculate the expected number of tokens being generated in a step of vertical cascade with $\mathcal{M}_{d_1}, \mathcal{M}_{d_2}$. With the property of generating function, the coefficient of term $x^j$ of $\phi^n(x)$ is the probability of the sum of acceptance length of $n$ speculative step being $j$. Therefore, $\phi^n(x)$ represents the probability generating function right before $\mathcal{M}_t$ performs the generation.

To achieve the expected number of token acceptances of a probability-generating function, we seek for an operator that can map the probability-generating function into the desired expectation.

To achieve the operator, we begin with a single polynomial term of $x^j$. Fortunately, given the end result of $\frac{1-\alpha^{j+1}}{(1-\alpha)}$ [14], the operator $T_\alpha(f(x)) = \frac{1-\alpha f(\alpha)}{(1-\alpha)}$ will convert $x^j$ to $\frac{1-\alpha^{j+1}}{(1-\alpha)}$. In addition, due to the linearity of the operator, this can be extended to any polynomial. Therefore, we achieved the desired operator to map a probability-generating function into the desired expectation.

Apply operator $T_\alpha$ to $\phi^n(x)$, we achieved the result of $\frac{1-\alpha\phi^n(\alpha)}{(1-\alpha)}$ for the expected number of accepted tokens. Furthermore, since the number of $\mathcal{M}_{d_1}$ calls is $n$, and $\mathcal{M}_{d_2}$ is called $k$ time for each $\mathcal{M}_{d_1}$ call given a total of $nk$ calls of $\mathcal{M}_{d_2}$. The time cost is $1 + nc_{d_1} + nkc_{d_2}$ which implied the EWIF of the system being $\frac{1-\alpha\phi^n(\alpha)}{(1-\alpha)(1+nc_{d_1}+nkc_{d_2})}$.

For Corollary 4.4, since both $0 < \alpha < 1$ and $0 < \alpha' < 1$, we have $\alpha^{i+1}\alpha'^{i+1} < \alpha\alpha'$, meaning that $\frac{1-\alpha^{k+1}\alpha'^{k+1}}{1-\alpha\alpha'} < 1$. Together with $\alpha - 1 < 0$, we have $\phi_{\alpha',k}(\alpha) < 1 + (\alpha - 1) = \alpha$. If we also let $nkc_{d_2} = 0$, we have $\frac{1-\alpha\phi^n(\alpha)}{(1-\alpha)(1+nc_{d_1}+nkc_{d_2})} > \frac{1-\alpha\alpha^n}{(1-\alpha)(1+nc_{d_1})}$, which is the EWIF for speculative decoding with step size $n$.

$\square$

