# OpenReview forum: "Cascade Speculative Drafting for Even Faster LLM Inference"
_NeurIPS.cc/2024/Conference — NeurIPS 2024 poster_

### Official Review · Reviewer_qvRP · 2024-06-18

**Soundness:** 3
**Presentation:** 2
**Contribution:** 3
**Rating:** 5
**Confidence:** 4

**Summary:**

This paper concentrates on the inference efficiency of LLMs and thinks that the autoregressive generation contained in drafting process of speculative decoding leads to the suboptimal performance of speculative decoding.  It introduces a novel speculative execution algorithm,  Cascade Speculative Drafting (CS Drafting), which containes Vertical Cascade and Horizontal Cascade to achieve better speedup, while preserving the same output distribution as the target model.

**Strengths:**

1. The proposed model introduces a new framework that fully considers the acceleration at different granularities and varying capabilities of the model to complete decoding.
2. The method proposed in this paper is model-agnostic and does not require additional training of the model.
3. Experimental results indicate that compared to the vanilla model, the proposed model can achieve better speedups.

**Weaknesses:**

1. The model employs a heuristic approach, involving a large number of hyperparameters, especially the $k$-matrix, whose quantity is related to the involved drift models.
2. The experimental results are not sufficient. On one hand, it does not compare with the latest speculative decoding methods. On the other hand, the model does not explore performance with different number of candidate tokens. It also lacks ablation studies and does not individually investigate the different impacts of the two cascade methods.
3. There are writing issues in the article, such as the first line of Algorithm 1, line 178, and line 292. Additionally, many variables in Algorithm 1 are not explained.

**Questions:**

1. The description of the core algorithm, Algorithm 1, in this paper is not clear. It would be helpful to include an explanation of the overall logic of the algorithm and add descriptions of some variables.
2. The paper lacks relevant ablation experiments to illustrate the relationship between the two cascade methods and lacks analytical experiments to demonstrate the robustness of the method with different numbers of candidate tokens.

**Limitations:**

The authors adequately addressed the limitations and, if applicable, potential negative societal impact of their work.

---

> ### Author Rebuttal · Authors · 2024-08-07
>
> We thank the reviewer for highlighting the advantages of our method. We appreciate the insightful and detailed feedback, and would like to address each of your concerns.
>
> > W1: The model employs a heuristic approach, involving a large number of hyperparameters, especially the K-matrix, whose quantity is related to the involved drift models.
>
>  We understand your concern about the number of hyperparameters. However, having additional hyperparameters is an intrinsic problem of algorithms utilizing speculative execution as hyperparameters are usually necessary to be adjusted based on the probability of acceptance.
>
> As a remedy to the additional hyperparameters, during our experiments with Vicuna-7B, we used the same hyperparameter across different datasets for each model. This represents the performance of CS Drafting without meticulous hyperparameter tuning.
>
> We also include the experiment results with different $K_{00}$ (the entry limiting the generation length of the largest draft model. This is the hyperparameter that has the largest effect on our experiments) on GSM8K using Vicuna-7B:
>
>
> | $K_{00}$ | Walltime (tokens/s) |
> | -------- | ------- |
> | 1  | 56.16 |
> | 2  | 55.51 |
> | 3 | 53.73 |
>
>
> As shown in our experimental results, the performance of CS Drafting only decreases slightly when the hyperparameter is sub-optimal. Therefore, the end user who does not require maximum performance can use a simple setup to achieve near-optimal performance with CS Drafting.
>
>
> >  W2: The experimental results are not sufficient. On one hand, it does not compare with the latest speculative decoding methods. On the other hand, the model does not explore performance with different number of candidate tokens. It also lacks ablation studies and does not individually investigate the different impacts of the two cascade methods.
>
> Thank you for your comment! We have included extensive results for different settings in Table 2 and have conducted a different set of experiments shown in Table 3. Following your suggestion, we include additional experimental results with different $ K_{00} $ (refer to response to W1 above).
>
> As an ablation study, we removed the horizontal cascade from CS Drafting. This results in a performance decrease from 56.16 to 53.55 when $ K_{00} = 1$. This demonstrates the effectiveness of the horizontal cascade. For the vertical cascade, the reviewer may refer to our results in Table 2. The performance usually decreases when we remove a draft model, indicating the effectiveness of the vertical cascade. Additionally, our theoretical analysis in Section 4 also supports the improvement of both cascades when isolating one of them.
>
> Regarding the comparison with other speculative decoding methods, we have compared our results with both Medusa and speculative decoding. While we are aware of newer methods such as Eagle or Hydra, we would like to remind you that our experiments were performed before their introduction and our preprints were available months earlier than theirs.
>
>
> >  W3: There are writing issues in the article, such as the first line of Algorithm 1, line 178, and line 292. Additionally, many variables in Algorithm 1 are not explained.
>
> Thank you for the detailed feedback on writing. We will do a more careful proofread to fix typos and add explanations of the variables.
>
> >  Q1: The description of the core algorithm, Algorithm 1, in this paper is not clear. It would be helpful to include an explanation of the overall logic of the algorithm and add descriptions of some variables.
>
> We would like to adopt the suggestion to improve the clarity. The following is the overview of our algorithm.
>
> The heart of the CS Drafting algorithm involves using smaller models as drafters for larger draft models (vertical cascade) as well as allocating smaller draft models to continue drafting less important tokens after larger draft models (horizontal cascade).
> Algorithm 1 implements these two cascades by using a for loop and recursion. The for loop is responsible for the horizontal cascade by gradually limiting the usable draft models to smaller models. The recursion calls smaller draft models to perform drafting for the target model or large draft models.
>
> We will also ensure that all variables in the algorithm are properly defined and are easy to understand to the readers.
>
> > Q2: The paper lacks relevant ablation experiments to illustrate the relationship between the two cascade methods and lacks analytical experiments to demonstrate the robustness of the method with different numbers of candidate tokens.
>
> Thank you for your suggestion! Please refer to our response to W1 and W2.
>
> Thank you again for reviewing our paper! We hope our response addresses your questions. Please let us know your thoughts, and we are more than happy to answer any further questions.

---

> > ### Comment · Reviewer_qvRP · 2024-08-12
> > **Response to Authors**
> >
> > Thank you very much for your response. After reading your reply, I am very pleased with the improvements in the score. I hope you can incorporate our suggestions into the final version.

---

> > > ### Author Response · Authors · 2024-08-12
> > >
> > > Thank you for your feedback! We are glad that our response addressed your concerns. We will incorporate your suggestions into the final version.

---

### Official Review · Reviewer_FGcM · 2024-07-11

**Soundness:** 3
**Presentation:** 3
**Contribution:** 3
**Rating:** 6
**Confidence:** 3

**Summary:**

The paper proposes to use multiple draft model for speculative decoding. Specifically, the smallest model could be the statistic language model which has negligible latency therefore reducing the cost of autoregressive regression. Experiments show the proposed method works better than baselines.

**Strengths:**

(1) The idea is very interesting and supported by analysis such as Figure 2.

(2) Experiments show the proposed method works well.

**Weaknesses:**

(1) Algorithm 1 is heavy and should be simplified for the reader.

(2) The whole pipeline seems to introduce lots of engineering work. For instance, we need to set hyperparameters for how to choose model size for each token position. Lenience is also used which is good but increases the complexity of the pipeline.

**Questions:**

N/A

---

> ### Author Rebuttal · Authors · 2024-08-07
>
> We thank the reviewer for highlighting the idea, analysis, and experiments of our paper. We appreciate the feedback provided and would like to address the weaknesses mentioned.
>
> > Q1: Algorithm 1 is heavy and should be simplified for the reader.
>
> While we made numerous attempts to simplify the algorithm, we always found that simplified versions missed essential details needed by end users. To better help readers understand our algorithm, we would like to add the following summary as a caption to the algorithm:
>
> The heart of the CS Drafting algorithm involves using smaller models as drafters for larger draft models (vertical cascade) as well as allocating smaller draft models to continue drafting less important tokens after larger draft models (horizontal cascade).
> Algorithm 1 implements these two cascades by using a for loop and recursion. The for loop is responsible for the horizontal cascade by gradually limiting the usable draft models to smaller models. The recursion calls smaller draft models to perform drafting for the target model or large draft models.
>
> We will also add more explanations of the variables to make them easier for the readers to understand.
>
> > Q2: The whole pipeline seems to introduce lots of engineering work. For instance, we need to set hyperparameters for how to choose model size for each token position. Lenience is also used which is good but increases the complexity of the pipeline.
>
>
>  We understand your concern about the number of hyperparameters. However, having additional hyperparameters is an intrinsic problem of algorithms utilizing speculative execution as hyperparameters are usually necessary to be adjusted based on the probability of acceptance.
>
> As a remedy to the additional hyperparameters, during our experiments with Vicuna-7B, we used the same hyperparameter across different datasets for each model. This represents the performance of CS Drafting without meticulous hyperparameter tuning.
>
> We also include the experiment results with different $K_{00}$ (the entry limiting the generation length of the largest draft model. This is the hyperparameter that has the largest effect on our experiments) on GSM8K using Vicuna-7B:
>
>
> | $K_{00}$ | Walltime (tokens/s) |
> | -------- | ------- |
> | 1  | 56.16 |
> | 2  | 55.51 |
> | 3 | 53.73 |
>
>
> As shown in our experimental results, the performance of CS Drafting only decreases slightly when the hyperparameter is sub-optimal. Therefore, the end user who does not require maximum performance can use a simple setup to achieve near-optimal performance with CS Drafting.

---

> > ### Author Response · Authors · 2024-08-12
> >
> > Dear Reviewer FGcM, thank you again for reviewing our paper! We hope our response addresses your questions. Please let us know your thoughts, and we are more than happy to answer any further questions.

---

### Official Review · Reviewer_LRu7 · 2024-07-13

**Soundness:** 3
**Presentation:** 2
**Contribution:** 3
**Rating:** 6
**Confidence:** 3

**Summary:**

This study introduces a novel method to accelerate large language model (LLM) decoding by integrating speculative decoding with two types of model cascades: vertical and horizontal. The horizontal cascade utilizes larger draft models for generating initial tokens, while smaller models assist in producing subsequent tokens. The vertical cascade implements a series of verification steps using a stack of cascaded models of varying sizes. Through the integration of these approaches, the authors report notable improvements in decoding speed across benchmarks.

**Strengths:**

**Originality** This paper demonstrates originality by expanding on speculative decoding. It introduces two novel techniques - horizontal and vertical cascades - that effectively factorize the draft-and-verification steps of speculative decoding across multiple models.

**Quality and Clarity** The authors base their approach in intuitive assumptions, like the complexity of first token generation. The speed improvements over vanilla speculative decoding illustrate the effectiveness of the proposed methods.

**Significance** The paper makes a considerable contribution to the community. Although CS drafting alone underperforms compared to Medusa (the multiple decoding heads method), the authors successfully combined their technique with Medusa, achieving superior performance. This suggests that their technique could be useful when integrated with other decoding methods.

**Weaknesses:**

**Algorithmic Complexity**: The paper discusses the use of horizontal and vertical cascades, but each additional cascade increases the algorithmic complexity of the decoding process. This complexity can become particularly challenging with larger models. The paper does not address how to balance this increased complexity with the potential speedup gains. Important questions such as the optimal number of cascades and the comparative usefulness of horizontal versus vertical cascades remain unanswered.

**Ablation Study**: The max-gram drafting technique appears to be quite efficient for generating many tokens without needing to invoke other medium-sized models. However, the paper lacks an ablation study to analyze the impact of different model combinations on decoding performance. Including such a study would provide valuable insights into the efficiency and effectiveness of the proposed method.

**Memory and Compute Constraints** The use of multiple models in complex scenarios poses challenges related to memory and computational limits. With limited memory and bandwidth, adding more cascades might not lead to faster end-to-end latency, especially as the target models grow in size. The paper does not consider this factor or provide a discussion on its implications.

**Questions:**

Please see the "Weaknesses" section.

**Limitations:**

Ok.

---

> ### Author Rebuttal · Authors · 2024-08-07
>
> We thank the reviewer for recognizing the significance of our contribution. We appreciate the insightful feedback provided and would like to address each of your questions.
>
> > Q1: Algorithmic Complexity
>
> To make an additional smaller draft model meaningful in the CS Drafting algorithm, it needs to be much smaller than the smallest used draft model, typically by a factor of at least 10. This represents an exponential relationship between the number of draft models and the size of the target model, so the total number of draft models should ideally be capped at 6, even for trillion-parameter models.
>
> Unfortunately, we do not have sufficient GPU capacity to conduct inference with models large enough to establish the relationship between the number of parameters and the ideal number of draft models. While it’s unclear how much additional performance gain can be achieved by further increasing the number of draft models, we expect a similar performance boost by our algorithm on larger models while keeping the same level of system complexity.
>
>
>
>
>
> > Q2: Ablation Study
>
> We have provided a thorough analysis on different combinations in Table 2. We are also happy to share the performance of various combinations of draft models for Vicuna-7B on GSM8K below:
>
> | Draft model | Walltime (Tokens/s) |
> | --- | --- |
> | MaG  |  52.89 |
> | Vicuna-68m | 44.31  |
> | Vicuna-68m & MaG | **56.72** |
>
>
> We observe the effectiveness of MaG as it exceeds the performance of Vicuna-68m due to its nearly zero latency. The optimal performance in our experiment is achieved by using both MaG and Vicuna-68m as draft models.
>
>
> > Q3: Memory and Compute Constraints
>
> We conducted an experiment to monitor memory usage across different systems accelerating Vicuna-7B on GSM8K:
>
> | Method | GPU Memory(MiB) |
> | -------- | ------- |
> | Huggingface Generation  | 14,016 |
> | CS Drafting | 16,154  |
> | Speculative Decoding | 16,118 |
>
> While both Speculative Decoding and CS Drafting utilize a moderate amount of memory compared to Huggingface generation, there is little difference between the performance of CS Drafting and Speculative decoding due to the additional draft models used by CS Drafting being much smaller than the first draft model.

---

> > ### Author Response · Authors · 2024-08-12
> >
> > Dear Reviewer LRu7, thank you again for reviewing our paper! We hope our response addresses your questions. Please let us know your thoughts, and we are more than happy to answer any further questions.

---

> > ### Comment · Reviewer_LRu7 · 2024-08-13
> >
> > Thank you to the authors for their response. It was helpful in gaining a better understanding of the paper. I am maintaining my positive score.

---

### Decision · Program_Chairs · 2024-09-25

**Decision:**

Accept (poster)

**Comment:**

This paper introduces a novel approach to accelerating LLM decoding by combining speculative decoding with vertical and horizontal model cascades. The originality is evident in the innovative techniques: horizontal cascades, which use larger models for initial tokens and smaller ones for subsequent tokens, and vertical cascades, which verify outputs through a series of cascaded models. The authors effectively demonstrate speed improvements over traditional speculative decoding and show that their method can enhance performance when integrated with existing techniques.

To further strengthen the paper, the authors should incorporate valuable discussions from the review period into the final version. This addition would address any remaining gaps and improve the clarity of the proposed methods. Overall, the paper is an original contribution to the field, and I recommend it for acceptance.